# Ganglionic Acetylcholine Receptor Antibodies and Autonomic Dysfunction in Autoimmune Rheumatic Diseases

**DOI:** 10.3390/ijms21041332

**Published:** 2020-02-16

**Authors:** Michie Imamura, Akihiro Mukaino, Koutaro Takamatsu, Hiroto Tsuboi, Osamu Higuchi, Hideki Nakamura, Saori Abe, Yukio Ando, Hidenori Matsuo, Tadashi Nakamura, Takayuki Sumida, Atsushi Kawakami, Shunya Nakane

**Affiliations:** 1Department of Neurology, Graduate School of Medical Sciences, Kumamoto University, Kumamoto 860-8556, Japan; kmichi1229@gmail.com (M.I.); takamakt@gmail.com (K.T.); andoy709@kumamoto-u.ac.jp (Y.A.); 2Department of Molecular Neurology and Therapeutics, Kumamoto University Hospital, Kumamoto 860-8556, Japan; a.mukaino@gmail.com; 3Department of Internal Medicine, Faculty of Medicine, University of Tsukuba, Ibaraki 305-8576, Japan; Hiroto-Tsuboi@md.tsukuba.ac.jp (H.T.); ori86114@gmail.com (S.A.); tsumida@md.tsukuba.ac.jp (T.S.); 4Department of Clinical Research, National Hospital Organization Nagasaki Kawatana Medical Center, Nagasaki 859-3615, Japan; osmhgc@gmail.com; 5Department of Immunology and Rheumatology, Division of Advanced Preventive Medical Sciences, Nagasaki University Graduate School of Medical Sciences, Nagasaki 852-8501, Japan; nhideki@nagasaki-u.ac.jp (H.N.); atsushik@nagasaki-u.ac.jp (A.K.); 6Department of Neurology, National Hospital Organization Nagasaki Kawatana Medical Center, Nagasaki 859-3615, Japan; matsuo.hidenori.wa@mail.hosp.go.jp; 7Department of Rheumatology, Kumamoto Sakurajyuji Hospital, Kumamoto 861-4173, Japan; t1.nakamura@sakurajyuji.jp

**Keywords:** autoimmune rheumatic diseases, ganglionic acetylcholine receptor antibody, autonomic neuropathy, autonomic dysfunction, Sjögren’s syndrome, systemic sclerosis, rheumatoid arthritis, systemic lupus erythematosus

## Abstract

Autonomic neuropathy has been reported in autoimmune rheumatic diseases (ARD) including Sjögren’s syndrome, systemic sclerosis, rheumatoid arthritis, and systemic lupus erythematosus. However, the pathophysiological mechanism underlying autonomic dysfunction remains unknown to researchers. On the other hand, autoimmune autonomic ganglionopathy (AAG) is an acquired immune-mediated disorder, which causes dysautonomia that is mediated by autoantibodies against ganglionic acetylcholine receptors (gAChRs). The purpose of this review was to describe the characteristics of autonomic disturbance through previous case reports and the functional tests used in these studies and address the importance of anti-gAChR antibodies. We have established luciferase immunoprecipitation systems to detect antibodies against gAChR in the past and determined the prevalence of gAChR antibodies in various autoimmune diseases including AAG and rheumatic diseases. Autonomic dysfunction, which affects lower parasympathetic and higher sympathetic activity, is usually observed in ARD. The anti-gAChR antibodies may play a crucial role in autonomic dysfunction observed in ARD. Further studies are necessary to determine whether anti-gAChR antibody levels are correlated with the severity of autonomic dysfunction in ARD.

## 1. Introduction

Autonomic neuropathy has been reported in autoimmune rheumatic diseases including Sjögren’s syndrome (SS) [1,2], systemic sclerosis (SSc) [3], rheumatoid arthritis (RA) [4], and systemic lupus erythematosus (SLE) [5]. Several underlying mechanisms such as the immunological basis, which includes circulating autoantibodies, abnormalities of cellular immunity, vasculitis, and secondary amyloidosis have been proposed.

Autoimmune autonomic ganglionopathy (AAG) is an acquired immune-mediated disorder that causes widespread autonomic failure, which is mediated by autoantibodies against the ganglionic acetylcholine receptor (gAChR) [6,7,8]. Clinical manifestations result from the impairment of sympathetic [orthostatic hypotension (OH) and anhidrosis] and parasympathetic activity (abnormal pupillary response, sexual dysfunction, and a fixed heart rate) [7,9]. The gAChRs located in autonomic ganglia in the sympathetic and parasympathetic nervous system have a pentameric structure, consisting of two α3 and three β4 subunits [10]. The Mayo Clinic group was the first to report that autoantibodies targeting gAChRs detected in the sera of approximately 50% of patients with idiopathic autonomic neuropathy were proven to be pathogenic [7,11]. These autoantibodies induce the internalization of cell-surface nicotinic gAChRs and subsequent impairment in synaptic transmission within the autonomic nervous system [12,13]. Radio-immunoprecipitation assay with [^125^I]-labeled epibatidine, which was developed by the Mayo Clinic, has been used to detect gAChR antibodies [6,7,8]. We developed luciferase immunoprecipitation systems (LIPS) to detect antibodies that specifically bind to the α3 or β4 gAChR subunits with high sensitivity. We performed the LIPS analysis with the α3 or β4 subunit fused to Gaussia Luciferase^8990^ to measure the respective antibodies in human sera [9,14]. Recently, we determined the prevalence of anti-gAChR antibodies in autoimmune rheumatic diseases (ARD) including SS, SSc, RA, and SLE with this method [15,16,17]. In the present review, we aimed to critically examine the current literature on autonomic neuropathy and autonomic function tests and to propose that anti-gAChR antibodies provide a new perspective on the mechanism underlying autonomic dysfunction in ARD.

We searched for previous reports of autonomic dysfunction associated with ARD using a PubMed search. Search terms used were “Autonomic dysfunction”, “Autonomic neuropathy”, “Sjögren’s syndrome”, “Systemic sclerosis”, “Rheumatoid arthritis”, and “Systemic lupus erythematosus”. Results of the search were screened for related studies by applying inclusion and exclusion criteria to the full text of the related studies. Article type included research article, short communication, case series, case reports, literature review published between 1983 and 2019.

## 2. Autonomic Dysfunction in Sjögren’s Syndrome

SS is a systemic autoimmune disease characterized by exocrine impairment of the salivary and lacrimal glands, in addition to various extraglandular features. Exocrine glandular dysfunction is a pathognomonic feature of SS. Exocrine glandular function is highly regulated by the autonomic nervous system [18]. Cholinergic dysfunction may be independent of the inflammation and atrophy of the exocrine glands [19]. Newton et al. reported that autonomic symptoms, which were common among patients with primary SS, may contribute to the overall symptom burden and are linked with systemic disease activity [20]. Several studies have estimated dysfunction in both parasympathetic and sympathetic nerves or only parasympathetic nerves through cardiovascular autonomic reflex testing [21,22,23,24,25,26,27,28]. However, the use of heart rate variability (HRV) has resulted in contradictory findings [29,30,31,32,33,34]. The predominance of parasympathetic vagal modulation of cardiac function at rest, coupled with reduced cardiac baroreceptor control of heart rate and increased sympathetic vasoconstrictor activity in patients with primary SS has been reported using microneurography [35].

Autonomic neuropathy characterized by various autonomic features including orthostatic intolerance (OI), and gastrointestinal (GI), sudomotor, pupillary and genitourinal impairment in patients with SS may be the initial manifestation and before or after the appearance of the “sicca” symptoms, accompanied by symptoms other than dysautonomia including sensory disturbance [36,37,38,39,40,41,42,43,44,45,46,47,48,49,50,51,52,53,54,55]. It could be localized [41,47,53] or systemic in distribution. Although autonomic neuropathy is potentially immunoresponsive, immunotherapy with intravenous immunoglobulin (IVIg) may require repetitive, continuous, or adjunctive therapy with rituximab for sustained improvement [56]. Pathological findings at autopsy revealed a decrease in the number of neurons within the thoracic sympathetic ganglia [50].

The possible pathogenesis of autonomic dysfunction in SS involves immunological factors including direct T-cell attack or ischemia caused by vasculitis in the autonomic ganglia and peripheral autonomic nerves [50,57,58], inhibition of neuropeptide secretion from nerve endings induced by cytokines [59], immune complex-mediated inflammation, and formation of pathogenic autoantibodies against receptors relative to autonomic function [60]. Type-3 muscarinic acetylcholine receptors (M3Rs) have been thought to be autoantigens in SS [61,62,63,64,65]. Importantly, Antibodies against M3Rs interfere with M3R-mediated parasympathetic neurotransmission and inhibit salivary secretion [66], GI motility [67], and bladder detrusor muscle contraction [68]. Passive transfer of SS immunoglobulins or rabbit antibodies to the second extracellular loop of the M3R in mice caused overactive bladder [69]. Moreover, neutralization of anti-M3R antibodies using IVIg improved the bladder and GI symptoms [70].

## 3. Autonomic Dysfunction in Systemic Scleroderma

SSc is an autoimmune rheumatic disease characterized by inflammation, vascular injury, autoantibody production, and fibrosis of the skin and internal organs [71]. Neurologic involvement including autonomic dysfunction has been recognized and reported in scleroderma [3,72,73,74,75,76]. Some manifestations of SSc including GI dysfunction [77] and impairment of microcirculation [78] are attributed to autonomic dysfunction [3]. Reduced vagal and increased sympathetic modulation at rest and deranged sympathetic response to orthostatic stress have been demonstrated by HRV analysis [79]. Cardiac autonomic dysfunction, which is related to right ventricular dysfunction [80], dysregulation of myocardial blood flow [81] and arrhythmic complications and mortality in patients with SSc [78] and precedes the development of fibrosis [82]. Autonomic dysfunction is also correlated with anorectal motility disorders [83], gastric compliance [84], and esophageal motor dysfunction [85,86]. Patients with more severe GI disease had significantly higher scores across several composite autonomic symptom scale-31 (COMPASS-31) subdomains, including OI and secretomotor dysfunction [87]. Pupillary autonomic dysfunction, including several patients with Adie’ tonic pupil [85,88,89,90], abnormal sympathetic skin response [91,92,93] and bladder dysfunction [94], was described. The pathogenesis of autonomic dysfunction in SSc remains elusive to researchers. However, it is thought to be caused by autoimmune damage to nerves, vascular disease, or direct nerve compression from tissue fibrosis [75]. Pulse cyclophosphamide therapy was reported to decrease sympathetic overdrive in a patient with early SSc [95].

It has been recognized that humoral immunity dysregulation plays an important role in SSc pathogenesis, and several antibodies can be detected in the sera of patients with SSc [96]. Three antibodies e.g., the anti-centromere antibody, the anti-scleroderma-70 antibody, and the anti- RNA polymerase III-70 antibody are specific for SSc and serve as specific markers [96]. Previous studies have demonstrated that GI dysmotility in SSc was associated with circulating autoantibodies against muscarinic AChRs and myenteric neurons [96,97,98,99,100,101,102,103]. Recently, McMahan and colleagues reported that patients with SSc and anti-RNPC3 antibodies had moderate-to-severe GI disease [104].

## 4. Autonomic Dysfunction in Rheumatoid Arthritis

RA is a chronic and systemic inflammatory condition that mainly affects the synovial joints but also presents with extra-articular manifestations [105]. Patients with RA present with autonomic symptoms including alterations of the skin, nails and hair, cyanotic extremities, peripheral vasospasm, tachycardia, and OH. Impaired heat rate responses to deep breathing, orthostasis and Valsalva maneuver (performed clinically for cardiovascular examination), abnormal HRV indices, impaired sympathetic skin response, and pupillary autonomic dysfunction have been demonstrated in a majority of patients with RA [106,107,108,109,110,111,112,113,114,115,116,117]. The prevalence of autonomic dysfunction detected by abnormal cardiovascular reflex tests varied from 33% to 86% (median prevalence: 60%) [4]. The characteristics of autonomic dysfunction (low HRV, reduced parasympathetic activity, elevated sympathetic activity and reduced cardiac baroreflex sensitivity [116]) are related to an increased risk of cardiovascular disease and mortality in patients with RA [4]. Autonomic dysfunction occurs in the early stage of RA and is not always a result of long-term RA and the inflammatory burden [117,118]. The association between the autonomic nervous system and elevated inflammation has been demonstrated using clinical cardiovascular tests and HRV [117,119,120,121]. Reduced parasympathetic and increased sympathetic activity on HRV were predictors of poor response to anti-tumor necrosis factor therapy [122]. The pathogenesis of autonomic dysfunction in patients with RA remains to be clearly elucidated in the literature. The presence of circulating autoantibodies against the nerve growth factor [123], cervical ganglia, and vagus nerve [124] and vasculitis of the vasa nervorum [125,126] and secondary amyloidosis have been proposed as the possible pathogenic mechanism. Proinflammatory cytokines have also been to linked to autonomic dysfunction. Cerebrospinal fluid interleukin-1β was inversely correlated with parasympathetic activity [127]. Interleukin-6 blockade with tocilizumab improved autonomic dysfunction in RA [125].

## 5. Autonomic Dysfunction in Systemic Lupus Erythematosus

SLE is an autoimmune disease that affects multiple organs and the central and peripheral nervous systems by the production of autoantibodies and immune-complex deposition [128]. The 1999 American College of Rheumatology provided definitions for seven peripheral manifestations (acute inflammatory demyelinating polyradiculopathy, autonomic disorder, mononeuropathy, myasthenia gravis, cranial neuropathy, plexopathy and polyneuropathy) related to SLE [129]. Abnormal HRV indices on the 24-h ambulatory electrocardiogram (ECG) and standard 12-lead ECG monitoring [130,131,132], abnormal sympathetic skin response [133], and pupillary autonomic disturbance [88] have been detected in patients with SLE. The prevalence of autonomic dysfunction ranged widely from 6 to 93% in patients with SLE [113,130]. Autonomic dysfunction in SLE is often asymptomatic and is not associated with disease activity, disease damage, and serological markers [131,134]. The findings of most of the studies reviewed supported sympathetic nervous system predominance or parasympathetic nervous system dysregulation, as reflected by decreased HRV in patients with SLE [135]. This autonomic imbalance is related to an increased risk of developing cardiovascular disease, which is a major cause of morbidity and mortality in patients with SLE [136,137]. Autonomic neuropathy can occur before or after the other clinical manifestations of SLE [138,139,140]. The precise pathogenic mechanism underlying autonomic dysfunction in SLE remains unclear. The positive response of several patients to immunotherapy for autonomic neuropathy characterized by severe sympathetic and parasympathetic impairment, supports the immune component in the pathogenesis of autonomic dysfunction in SLE [138,141,142]. Circulating autoantibodies against the nerve growth factor [123], cervical ganglia and vagus nerve (similar to rheumatoid arthritis) has been proposed [124].

## 6. Anti-Ganglionic Acetylcholine Receptor Antibody in Autoimmune Rheumatic Disease

### 6.1. Case Series and Prevalence of Anti-gAChR Antibodies in Sjögren’s Syndrome

Kondo et al. described two Japanese patients with SS, who presented with chronically progressive dysautonomia. One patient was seropositive for anti-gAChR antibodies and improved after oral intake of prednisolone. This case indicated that anti-gAChR antibodies are relevant to the pathophysiology of SS [143]. We also described a seropositive patient with SS, who presented with recurrent abdominal distention, constipation, weight loss, orthostatic dizziness, anhidrosis, and incomplete bladder emptying [144].

We determined the associations between autonomic dysfunction and anti-gAChR antibodies in SS by using the LIPS assay. We found that 9/39 (23.1%) patients with SS were seropositive and five of nine patients had autonomic symptoms. Moreover, we detected anti-gAChRα3 and anti-gAChRβ4 antibodies in 8/10 (80.0%) patients with SS with autonomic symptoms [15]. To identify the clinical characteristics of primary SS in patients with the gAChR antibodies, we obtained 22 serum samples (from 22 patients with primary SS) from teaching and general hospitals throughout Japan between January 2012 and March 2017 [145]. Clinical diagnoses were made in each hospital, and the diagnosis of SS was confirmed using the diagnostic criteria proposed by the American European Consensus Group and/or the Japanese Ministry of Health criteria for SS diagnostics. We measured levels of autoantibodies against gAChRα3 and gAChRβ4. A total of 11/22 patients tested positive for antibodies, including 9 gAChRα3-positive and 2 double antibody-positive patients. We reviewed clinical features and laboratory data (cerebrospinal fluid findings, other antibodies, etc.) for these 22 patients with primary SS (mean age: 53 years old; 6 men and 16 women). Chronic autonomic dysfunction was the predominant subtype in our investigation (6/16, 38%). No significant differences in clinical features involving autonomic dysfunction and laboratory data were noted, except for a higher frequency of OH/OI in the patients who were positive for the anti-gAChR antibodies compared to the antibody-negative patients (90% vs. 44%, *p* = 0.043).

### 6.2. Case Series and Prevalence of Anti-gAChR Antibodies in Systemic Sclerosis, Systemic Lupus Erythematosus, and Rheumatoid Arthritis

We described the cases of three patients with ARD and AAG. Patient 1 demonstrated an overlap between RA and SS, patient 2 demonstrated an overlap between RA and SSc, and patient 3 demonstrated an overlap between RA and SLE. Anti-gAChRα3 antibodies were detected in patients 1 and 2. A variety of autonomic nervous symptoms such as OI, early satiety, constipation, and diarrhea were observed. All patients received steroid pulse therapy, and their clinical symptoms improved [146].

We determined the associations between autonomic dysfunction and anti-gAChR antibodies in SSc, SLE, and RA using the LIPS assay. The LIPS assay detected anti-gAChRα3 and β4 antibodies in the sera of patients with SSc (13.2%, 5/38), SLE (12.5%, 4/32), and RA (18.6%, 8/43) [17].

We investigated human leukocyte antigen (HLA) alleles in patients with autoimmune hepatitis with or without anti-gAChR antibodies earlier. The frequency of the HLA-DRB1*0403 allele differed among Japanese patients with autoimmune hepatitis, based on the presence or absence of anti-gAChR antibodies. Thus, we should confirm the association of the HLA allele and each ARD [16].

### 6.3. Illustrative Cases That Tested Positive for Anti-gAChR Autoantibodies

#### 6.3.1. Patient 1: SS with Widespread Neurological Symptoms Including Autonomic Dysfunction

A 58-year-old woman presented with a wide range of autonomic symptoms including sicca symptoms, constipation, OH and anhidrosis, pupillary abnormality two years ago. Neurological examination revealed “glove and stocking“ paresthesia, pyramidal tract sign, and ataxia. Although the patient tested negative for anti-SSA and SSB antibodies, lip biopsy demonstrated the infiltration of inflammatory cells around the ducts. She was diagnosed with primary SS, according to the American-European consensus criteria. She tested positive for anti-gAChRα3 and anti-M3R antibodies. The level of anti-gAChRα3 antibodies detected in serum was 1.343 antibody index (AI) (normal value < 1.000) using the LIPS assay. The serum levels of the second extracellular domain of M3R antibodies and the N-terminal of M3R antibodies detected by ELISA were 0.203 (normal value < 0.103) and 0.185 (normal value < 0.074), respectively. The coefficient of variation in R-R intervals (CVRR) was normal (4.15%, normal value > 1.4), but the coefficient of low frequency/high frequency was very low (0.35, normal range = 0.8–2.0) (Table 1). Sudomotor function testing revealed sympathetic dysfunction (Figure 1). She was treated with intravenous methylprednisolone (IVMP) (methylprednisolone 1 g per day for 3 days), which was followed by IVIg therapy (15 g per day for 5 days) twice in two months, based on the clinical diagnosis of SS with neurological symptoms including widespread autonomic dysfunction. She demonstrated improvement in OH and experienced recovery of sweating function with IVMP followed by IVIg. The levels of the anti-gAChR and anti-M3R antibodies returned to normal after immunotherapy, which resulted in improvement on autonomic function examination (Table 1 and Figure 1). Especially, we confirmed the denervation supersensitivity was improved in autonomic testing for pupil abnormality.

Anhidrosis was confirmed before immunotherapy, except on the axillary and palmar surfaces. Patient 1 was administered intravenous methylprednisolone, followed by monthly infusions of intravenous immunoglobulin for two months. Consequently, the anhidrosis improved with combination therapy, accompanied by a decrease in serum anti-gAChR and anti-M3R antibody levels.

#### 6.3.2. Patient 2: Systemic Scleroderma and Autonomic Manifestations

A 76-year-old woman who had SSc for at least 9 years also experienced OH/OI and alternate bowel habits. She experienced recurrent light-headedness upon standing, severe constipation, and diarrhoea. Ocular examination confirmed a pupillary abnormality in the left eye. The serum level of anti-gAChRα3 antibodies was 3.607 AI. The CVRR was low (0.72%, normal value > 2.3). Moreover, the plasma norepinephrine (NE) level was also low in this patient (21, normal range = 90–420 pg/mL). Treatment with two doses of IVMP, followed by administration of oral prednisolone (1 mg/kg of body weight) based on the clinical diagnosis of SSc with AAG, led to sustained improvement in the autonomic symptoms. Subsequently, oral administration of tacrolimus was initiated at a dose of 3 mg once daily after obtaining informed consent. She maintained improvements in OI and experienced recovery of GI function with IVMP therapy, which was followed by oral prednisolone and tacrolimus treatment. The levels of anti-gAChR antibodies in this patient decreased after treatment (α3, 1.445 AI), which resulted in an improvement in CVRR (1.26%) and plasma NE level (679 pg/mL), respectively.

## 7. Limitations

This review has some limitations. First, the present study included a small study population. Further studies are necessary to determine the prevalence of autonomic disturbances in ARD. Second, we focused on only one aspect of the autonomic disturbances in of ARD, i.e., anti-gAChR antibodies. We should examine the immunological and autonomic biomarkers related to autonomic disturbance that could predict response to treatment. Third, a direct association between autonomic disturbance in ARD and gAChR antibodies has still not been proven. We should investigate the prevalence of autonomic disturbance and the immunopathogenesis of anti-gAChR antibodies for each rheumatic disease.

## 8. Conclusions

We performed a literature review of studies on autonomic dysfunction and determined that the prevalence of anti-gAChR antibodies was significant in ARD. Autonomic dysfunction, which is composed of parasympathetic underactivity and sympathetic overdrive is usually observed in ARD. Anti-gAChR antibodies may play a crucial role in autonomic dysfunction in ARD (Figure 2). Several phenomena other than anti-gAChR antibodies may play overlapping roles in the development of autonomic symptoms in ARD, and immune dysregulation of the autonomic nervous system could be responsible for the pathogenesis of ARD. Further studies are necessary to determine whether anti-gAChR antibody levels are correlated with the severity of autonomic dysfunction in ARD and to confirm the treatment strategy for countering autonomic dysfunction in ARD. Finally, rheumatologists and neurologists should foster cooperation among physicians, share clinical experiences and promote translational research. Moreover, we should expand on the concept and significance of our research, so that non-specialists can understand it and find it interesting.

Autonomic manifestations are frequently observed in patients with ARD, i.e., Sjögren’s syndrome (SS), systemic sclerosis (SSc), rheumatoid arthritis (RA), and systemic lupus erythematosus (SLE). It is well known that anti-M3R autoantibodies cause autonomic dysfunction, including the sicca complex and gastrointestinal dysmotility in ARD. On the other hand, anti-gAChR autoantibodies have the potential to physiologically block ganglionic synaptic transmission in patients with autoimmune autonomic ganglionopathy (AAG). Autoimmune diseases were observed in approximately 30% of AAGs in our previous study. We consider that AAG and other autoimmune diseases can coexist, owing to the same autoimmune basis. There is the possibility that the anti-gAChR autoantibodies contribute to the autonomic manifestations associated with ARD. We recommend that the highest priority for future experimental work in this field should be to analyze the autoantibodies against all the receptors in the autonomic nervous system such as M3R, gAChR, etc.

## Figures and Tables

**Figure 1 ijms-21-01332-f001:**
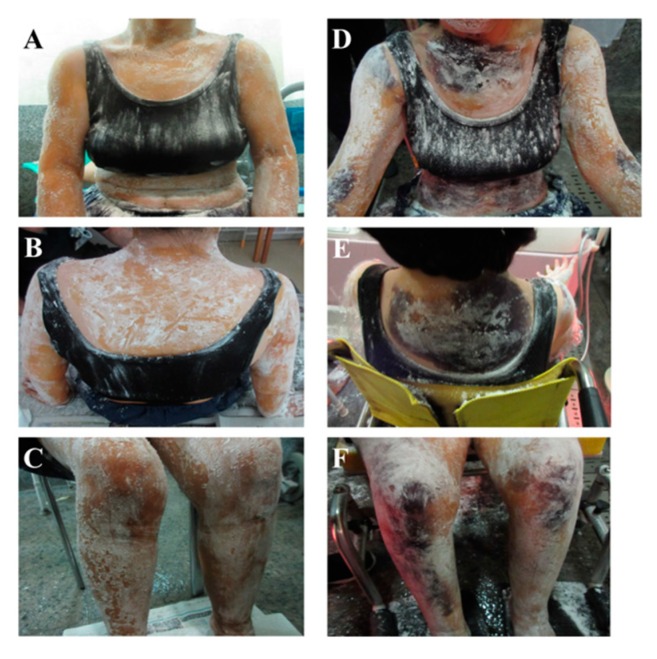
Thermoregulatory sweat test in Patient 1 (**A**–**C**): Pre-immunotherapy, (**D**–**F**): Post-immunotherapy).

**Figure 2 ijms-21-01332-f002:**
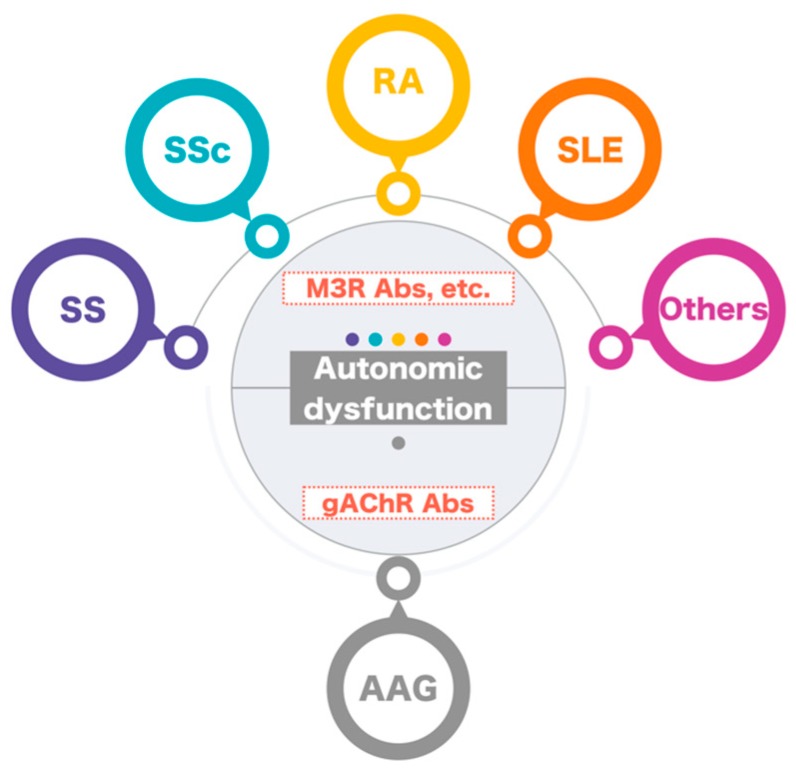
Anti-gAChR autoantibodies and autonomic dysfunction in autoimmune rheumatic diseases.

**Table 1 ijms-21-01332-t001:** Autonomic function tests in Patient 1.

	Pre-Immunotherapies	Post-Immunotherapies	Response to Immunotherapies
*Pupillary responses to local instillation*			
Before local instillation (mm, R/L)	5.0 */5.0 *	4.0/4.0	Improved
5% cocaine (mm, R/L)	6.0 */6.0 *	5.5/5.5	
1% phenylephrine (mm, R/L)	6.5 */5.5 *	4.0/4.0	
0.125% pilocarpine (mm, R/L)	2.5 */3.5	3.5/3.5	
*Secretomotor function test*			
Gum test (mL/10 min)	7.0 *	12.0	Improved
Schirmer tear test (mm/5 min)	5 */2 *	N.D. ^7^	–
*Sudomotor and cutaneous vasomotor test*			
Thermoregulatory sweat test	(Figure 1)		Improved
Acetylcholine sweat test	No response *	N.D.	–
*Cardiovascular function test*			
BP^1^ response to postural change, BP (mmHg)	171/101→143/87 *	145/95→144/90	Improved
HR^2^ response to postural change, HR (/min)	65→95	85→93	
CVRR ^3^ (%)	4.15	3.92	(W.N.L. ^8^)
LF/HF ^4^	0.35 *	0.20 *	Not effective
Myocardial ^123^I-MIBG ^5^ scintigraphy, H/M ratio ^6^ (early)	3.14	3.08	(W.N.L.)
Myocardial ^123^I-MIBG scintigraphy, H/M ratio (delayed)	3.82	4.04	(W.N.L.)

* Mean abnormal value from autonomic function test data. ^1^ BP, blood pressure; ^2^ HR, heart rate; ^3^ CVRR, coefficient of variation in R-R intervals; ^4^ LF/HF, low frequency/high frequency; ^5^ MIBG, metaiodobenzylguanidine; ^6^ H/M ratio, heart-to-mediastinum ratio; ^7^ N.D., not done; ^8^ W.N.L., within normal limits.

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
