# Peer review of "Ganglionic Acetylcholine Receptor Antibodies and Autonomic Dysfunction in Autoimmune Rheumatic Diseases"

_ijms, 2020, doi:10.3390/ijms21041332_

Round 1
Reviewer 1 Report
The study reports many previous results obtained from the analysis of the literature, together with results obtained by the participating groups of rheumatic medicine. In view of the combination of the previous and recent data, envisaged also by significant techniques, the paper is in general comprehensive, and therefore also potentially interesting. On the other hand, its strategical focus deals primarily with one aspect, the investigation of the ganglionic Ach receptor antibodies in the various rheumatic diseases. For this aspect and the other properties investigated, the data obtained in the various diseases are very heterogeneous. This concept is not new, it is only reinforced. No attempt is made for providing explanations. Rather all attempts, which are still undefined, are delayed to future studies. Based on these considerations the significance of this review is limited. In other words the review, although extensive, detailed and accurate, does not expand the knowledge and understanding of the diseases. My recommendation is to reconsider the work, expanding its significance which should be of interest also to non-specialist readers.
Author Response
Reviewer 1
The study reports many previous results obtained from the analysis of the literature, together with results obtained by the participating groups of rheumatic medicine. In view of the combination of the previous and recent data, envisaged also by significant techniques, the paper is in general comprehensive, and therefore also potentially interesting. On the other hand, its strategical focus deals primarily with one aspect, the investigation of the ganglionic Ach receptor antibodies in the various rheumatic diseases. For this aspect and the other properties investigated, the data obtained in the various diseases are very heterogeneous. This concept is not new, it is only reinforced. No attempt is made for providing explanations. Rather all attempts, which are still undefined, are delayed to future studies. Based on these considerations the significance of this review is limited. In other words the review, although extensive, detailed and accurate, does not expand the knowledge and understanding of the diseases. My recommendation is to reconsider the work, expanding its significance which should be of interest also to non-specialist readers.
Answer.
We would like to thank the reviewer for the helpful comment. As you mentioned above, our research address only one aspect, ganglionic AChR antibodies in autonomic disturbance of the various rheumatic diseases. We added ‘Limitations’ section (page 7/ lines 294-302).
“This review has some limitations. First, the present study included a small study population. Further studies are necessary to determine the prevalence of autonomic disturbances in ARD. Second, we focused on only one aspect of the autonomic disturbances in of ARD, i.e., anti-gAChR antibodies. We should examine the immunological and autonomic biomarkers related to autonomic disturbance that could predict response to treatment. Third, a direct association between autonomic disturbance in ARD and gAChR antibodies has still not been proven. We should investigate the prevalence of autonomic disturbance and the immunopathogenesis of anti-gAChR antibodies for each rheumatic disease.”
Moreover, we added the sentence (page 8/ lines 308-310).
“Several phenomena other than anti-gAChR antibodies may play overlapping roles in the development of autonomic symptoms in ARD, and immune dysregulation of the autonomic nervous system could be responsible for the pathogenesis of ARD.”
We would like to thank for your accurate advice that the work should be expanded its significance which should be of interest also to non-specialist readers. We have added the sentence (Page 8/ line 314-316).
“Moreover, we should expand on the concept and significance of our research, so that non-specialists can understand it and find it interesting.”
Reviewer 2 Report
The review is overall well written and interesting.
1) I suggest the Authors write the conclusions more precisely - what further directions in the treatments can be predicted. After finishing the last sentence one might have the impression that the paper and its conclusions are not yet finished.
2) It would benefit the readers if the Figure 2 was more described - all the disease names should be stated in the captions so that the Figure is clear by itself - it can be viewed on the MDPI website sometimes as a separate picture, so the caption should allow reading the message of the paper even without reading it.
Author Response
Reviewer 2
The review is overall well written and interesting.
1) I suggest the Authors write the conclusions more precisely - what further directions in the treatments can be predicted. After finishing the last sentence one might have the impression that the paper and its conclusions are not yet finished.
Answer.
We would like to thank the reviewer for the helpful comment. We modified the sentence (page 8/ lines 310-313).
“Further studies are necessary to determine whether anti-gAChR antibody levels are correlated with the severity of autonomic dysfunction in ARD and to confirm the treatment strategy for countering autonomic dysfunction in ARD.”
2) It would benefit the readers if the Figure 2 was more described - all the disease names should be stated in the captions so that the Figure is clear by itself - it can be viewed on the MDPI website sometimes as a separate picture, so the caption should allow reading the message of the paper even without reading it.
Answer.
Yes, we agree with the reviewer’s comment. In accord with the reviewer’s comment,we have rewritten Figure legend for Figure 2 in more detail(P8/ lines 321-332).
“Autonomic manifestations are frequently observed in patients with ARD, i.e., Sjögren’s syndrome (SS), systemic sclerosis (SSc), rheumatoid arthritis (RA), and systemic lupus erythematosus (SLE). It is well known that anti-M3R autoantibodies cause autonomic dysfunction, including the sicca complex and gastrointestinal dysmotility in ARD. On the other hand, anti-gAChR autoantibodies have the potential to physiologically block ganglionic synaptic transmission in patients with autoimmune autonomic ganglionopathy (AAG). Autoimmune diseases were observed in approximately 30% of AAGs in our previous study. We consider that AAG and other autoimmune diseases can coexist, owing to the same autoimmune basis. There is the possibility that the anti-gAChR autoantibodies contribute to the autonomic manifestations associated with ARD. We recommend that the highest priority for future experimental work in this field should be to analyze the autoantibodies against all the receptors in the autonomic nervous system such as M3R, gAChR, etc.”
Round 2
Reviewer 1 Report
The paper has been improved following many suggestion of the reviewers. As previously accepted by the authors, and now mentioned in the paper, the originality of this review is limited. On the other hand, the provided information about the various diseases is useful and could be relevant for the future development of appropriate therapy. As a whole, upon acceptance of some properties typical of the English language (for example, wider use of the commas), the publication appears reasonable.
Author Response
Yes, we agree with the reviewer’s comment. We modified the properties typical of the English language (especially, use of the commas and semicolons) (page 1/ lines 10, page 2/line 58-59, 93, page 3/line 109, 136, 142, page 4/line 158, 159, 165, 174, 183, 198, 245, page 8/line 314, 333). We changed the sentence (page 3/line 135-137) “Cardiac autonomic dysfunction, which is related to right ventricular dysfunction [82], dysregulation of myocardial blood flow [83],and arrhythmic complications and mortality [80], precedes the development of fibrosis in patients with SSc [84].”
Moreover, we modified some minor changes (page 3/line 123, page 5/line 224, 246, page 6/line 269, and almost reference section including page 12/line 483, page 13/line 496, 519, page 14/line 544, 557, page 15/line 596, and page 17/line 680, 692)